# A Parallel Processing Approach to Dynamic Simulation of Ethylbenzene Process

**Junkai Zhang** **, Zhongqi Liu, Zengzhi Du \* and Jianhong Wang**

Center for Process Simulation & Optimization, College of Chemical Engineering, Beijing University of Chemical Technology, Beijing 100029, China; jkzhang3@mail.buct.edu.cn (J.Z.); 2019200186@mail.buct.edu.cn (Z.L.); wjhmaster@263.net (J.W.)
\* Correspondence: duzz@mail.buct.edu.cn

**Abstract:** Parallel computing has been developed for many years in chemical process simulation. However, existing research on parallel computing in dynamic simulation cannot take full advantage of computer performance. More and more applications of data-driven methods and increasing complexity in chemical processes need faster dynamic simulators. In this research, we discuss the upper limit of speed-up for dynamic simulation of the chemical process. Then we design a parallel program considering the process model solving sequence and rewrite the General dynamic simulation & optimization system (DSO) with two levels of parallelism, multithreading parallelism and vectorized parallelism. The dependency between subtasks and the characteristic of the hottest subroutines are analyzed. Finally, the accelerating effect of the parallel simulator is tested based on a $500 \text{ kt} \cdot \text{a}^{-1}$ ethylbenzene process simulation. A 5-hour process simulation shows that the highest speed-up ratio to the original program is 261%, and the simulation finished in 70.98 s wall clock time.

**Keywords:** chemical process simulation; dynamic simulation; parallel computing; multithreading; vectorization

## 1. Introduction

Dynamic simulation is widely used by chemical engineers to better understand the process [1]. The simulator based on the first principles model has a wide range of applications on existing processes, and can also have certain predictions for some new processes, which has been proved to be effective in the past few decades. With the development of the fourth industrial revolution and the maturity of the big data environment, data-driven modeling becomes more widely accepted [2]. However, the corner case cannot be well modeled, such as start-up, shutdown, and fault if the data-driven elements are used only in modeling. In addition, the application of typical data-driven methods, such as model predictive control, optimal control, and reinforcement learning in the complex chemical process may be infeasible without the initial data provided by the first-principles modeling simulator in the way shown in Figure 1 [3]. Thus, the development of data-driven methods highly depends on the accuracy and speed of solving the first-principles modeling simulator. Modeling combining mechanistic and data-driven elements can reveal the character of the chemical process better.

For a single unit operation or small-scale process, Luyben [4] reported that the default model exported by Aspen plus cannot accurately predict the corner case because "the default heat-exchanger models do not account for heat-exchanger dynamics". Hecht et al. [5] further reported a similar problem in the reactor. For the plantwide scale, one of the most commonly used and discussed process simulators is the Tennessee Eastman process (TEP) simulator which is carried out by Downs and Vogel [6] based on the simulation model by the Eastman company. The model was built according to the real process but the components, kinetics, process, and operating conditions were modified. There are seven

components and five unit operations in the TEP model, and it also balances accuracy and difficulty in model solving, which can run at a satisfactory speed.

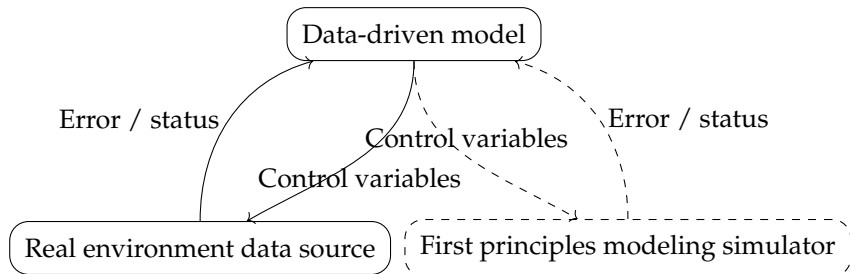

**Figure 1.** The relation between data-driven method and first-principles modeling simulator.

In recent years, chemical engineering processes have been experiencing an increase in complexity and scale. At the same time, the accuracy of modeling and solving such processes is definitely required. However, solving a more accurate mechanism model is often slower, especially for the large-scale process or multiple components. The efficiency of these process simulation is unacceptable for data generation. At the same time, the simulator must run faster than the real process and leave sufficient margin for optimization and control algorithms to meet the requirements of the application. Otherwise, it can not act as a predictor when combined with the real process. Thus, it is very important to implement an accurate and fast process simulator.

To balance the accuracy and difficulty of modeling and solving, model simplifying and solver accelerating have been tried. Sahlodin et al. [7] used non-smooth differential-algebraic equations (DAE) to model the dynamic phase change, which replaced the optimization problems solved by the original modeling method and reduced the time cost greatly. Connolly et al. [8] used a reduced variables method to simplify the hydrocarbon-water phase-equilibrium model. Li et al. [9] used a deep neural network model to simplify the solving of complex fluid mixtures NVT flash problem with given moles, volume, and temperature. These simplified models work well in the scope of simplification, but the previous full mechanism modeling cannot be used effectively.

In recent years, parallel computing has been widely used in chemical process simulation to realize the solver acceleration. Wang et al. [10] summarized that chemical process simulation and optimization could be divided into high-level and low-level parallelization, and discussed the problems of parallelization scale, load balance, and parallel efficiency. Washington and Swartz [11] solved the uncertain dynamic optimization problem using the direct multiple shooting method in parallel and applied it to design the chemical process. Laird et al. [12] also carried out a parallel solving process of dynamic DAE constrained optimization problems, which ran in parallel during the solving process of Karush–Kuhn–Tucker (KKT) system. These methods mainly run in parallel during optimization and dynamic integration, which are typical high-level parallelism, but model solving was not parallelized. Vegeais and Stadtherr [13], Mallaya et al. [14] reported that it was a typical low-level parallel algorithm to solve the efficiency problem of the linear solver by dividing the large sparse matrix into blocks. However, there are still a lot of non-linear models in the process of dynamic simulation, so there is still much room to improve the non-linear parallel solver. Chen et al. [15] realized coarse-grained and fine-grained parallelism on CPU and GPU. Further, Weng et al. [16] developed a dynamic simulation of molecular weight distribution of multisite polymerization with coarse-grained multithreading parallelism.

In the past few decades, Moore's law has predicted the growth of CPU single-core performance, and the speed of the same program also become faster without modification. However, the performance growth has encountered a bottleneck in recent years. More cores and vectorized instructions are added to the new CPU to maintain performance growth, which provides parallel computing capabilities on a single CPU chip. Unfortunately, parallel computing needs modifications on the program. At present, an effective solution

of increasing solving speed is to convert the problem into multiple sub-problems without dependence on each other and solve these sub-problems in parallel with the new CPU.

In this paper, we design a parallel dynamic simulation taking into account the character of process and the development of computer. Multithreading and vectorization parallel computing modifications are carried out based on General dynamic simulation & optimization system (DSO), which make full use of the features of modern CPU. Compared to the previous research, we use high-level multithreading parallelism and assign tasks according to unit operations which brings clearer task allocation and lower communication costs. The effect of parallel computing modifications is tested on a $500 \, \text{kt} \cdot \text{a}^{-1}$ ethylbenzene process simulator.

## 2. Process Dynamic Simulation

### 2.1. Current Program

For dynamic simulation, the solving of temperature, pressure, liquid level, concentration, and other parameters changing with time is an initial value problem of DAE with the constraints of the pipeline network. The form is given by Equation (1), where $\tau$ is time and $y$ are process variables.

$$\begin{cases} \dfrac{\mathrm{d}y}{\mathrm{d}\tau} = & f(\tau, y) \\ g(\tau, y) = & 0 \\ y(\tau_0) = & y_0 \end{cases} \tag{1}$$

For the chemical process, the improved Euler method is used to solve the problem. The iterative form is given by Equation (2), where $h$ is the integral step [17].

$$\begin{cases} y_{i+1} = & y_i + hf(\tau_{i+1}, y_{i+1}) \\ \tau_{i+1} = & \tau_i + h \\ y_0 = & y(\tau_0) \end{cases} \tag{2}$$

Due to the input of intermediate control variables during the dynamic simulation, the actual model form to be solved is given by Equation (3), where $c$ are input control variables.

$$\begin{cases} \dfrac{\mathrm{d}y}{\mathrm{d}\tau} = & f(\tau, y, c) \\ g(\tau, y) = & 0 \\ \phi(y, c) = & 0 \\ c = & c(\tau) \end{cases} \tag{3}$$

The existing diagram of solving the model in DSO is divided into four parts: solving unit operation model, pipeline network equations, control model, and numerical integration. The solving sequence is shown in Figure 2a.

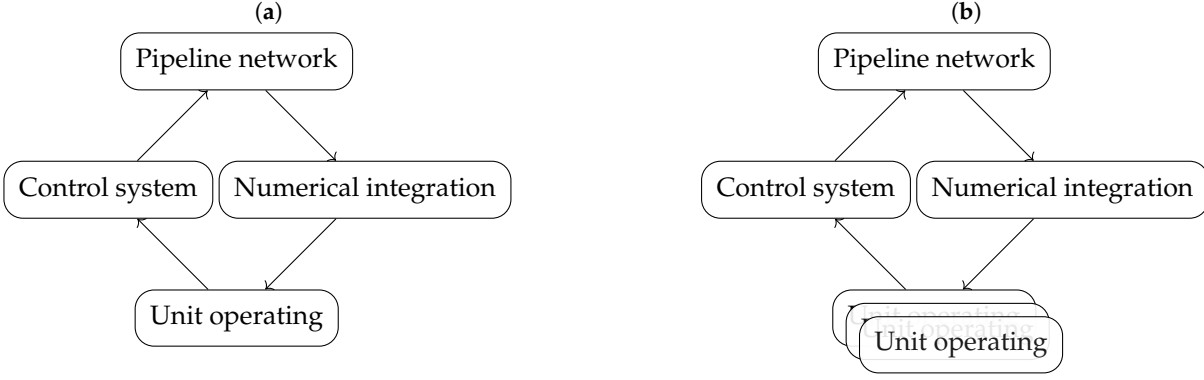

**Figure 2.** The diagram of solving the model in DSO. (**a**) Conventional serial sequence. (**b**) Parallel sequence.

## 2.2. The Choice of Parallel Level

Parallel processing means that the process of solving the problem is divided into several smaller parts and run on multiple processors at the same time to reduce the running time, and then the parts are combined to produce the final result. When there are dependencies in the problem pieces, the calculation process has to be carried out in serial. Partitioning, communication, synchronization, and load balancing are four typical considerations in the design of parallel programs [18].

The parallel part is designed according to the character of dynamic simulation. The numerical integration cannot be parallelized because it is mostly iterative and changes based on past iteration. During solving the unit operation model, the mass balance equation, heat balance equation, physical property, thermodynamic parameters method, and reactions are under consideration. As the short step size is selected, the calculation of a single unit operation in a one-step integral iteration can be seen as independent of other unit operations, and the numerical integration inside the unit operation is independent of other unit operations. During solving the control system model, there is dependence in the cascade controller in one-step iteration. The pipeline network model is dependent because it needs to be solved iteratively in the field of directly connected pipes.

First of all, from the perspective of partitioning in the design of the parallel program, hotspots are found according to the debugging call tree shown in Figure 3 and the most time-consuming function is listed in Table 1. We find that solving unit operation model takes about 75% of time in the whole program, and the most frequently called subroutines are the calculation of enthalpy, volume, and flash, which is close to the thermodynamic and physical property calculation time reported by Harrison [19]. The single unit operation model solving subroutines should be paralleled, and the thermodynamic and physical property calculation should also be optimized to reduce the hotspot time cost in solving the unit operation model.

**Table 1.** The most time-consuming function.

| Function Name | Inclusive Samples/% | Exclusive Samples/% |
| --- | --- | --- |
| sub_tray | 40.93% | 23.63% |
| enth | 6.94% | 6.87% |
| flash_b | 8.41% | 6.09% |
| round | 5.65% | 5.61% |
| subdtank | 12.67% | 5.58% |
| enth_to_t | 5.41% | 5.35% |
| kmoltokg | 4.03% | 3.93% |
| liquid_volume | 3.99% | 3.92% |
| sub_reactor | 5.06% | 3.38% |

As there are control variables given instantly by the operator or external program during simulation, the simulator is sensitive to delay. At the same time, the workload of the non-dependent parallelizable part is small. From the perspective of the communication and synchronization cost, the shared-memory system is selected. Considering the existing CPU specifications, especially the rapid growth of the number of cores, we choose shared-memory multi-threaded programming to modify the existing program. It avoids the high delay in offloading tasks to heterogeneous computing such as GPU and the difficulty of using network connections for multiple machines. Combining the problem characteristics and computer characteristics, the multithreaded parallel is selected to accelerate the unit operation model solving process as shown in Figure 2b.

Despite parallelizing the unit operation with multithreading, it is also necessary to rewrite the enthalpy, volume calculation, and flash result calculation functions because they are called for many times. As this calculation involves a lot of same calculations for each component, the vectorization parallel can speed up the calculation on the previous

basis by using the vector processors of modern CPU which can deal with multiple datasets at the same time, so the vectorization parallel is used to rewrite the calculation functions of enthalpy, volume, and flash results.

Therefore, we decided to use multithreaded parallel on solving unit operation model and vectorization parallel on thermodynamic and physical property calculation.

| Function name | Inclusive Samples/% | Exclusive Samples/% |
|---|---|---|
| + dynamic.exe | 100.00% | 0.00% |
| \| + [External code] | 99.04% | 0.37% |
| \| \| + WinMainCRTStartup | 98.65% | 0.00% |
| \| \| \| + __scrt_common_main | 98.65% | 0.00% |
| \| \| \| \| + __scrt_common_main_seh | 98.65% | 0.00% |
| \| \| \| \| \| + invoke_main | 98.65% | 0.00% |
| \| \| \| \| \| \| + WinMain | 98.65% | 0.00% |
| \| \| \| \| \| \| \| + [External call] DispatchMessageA | 98.63% | 0.01% |
| \| \| \| \| \| \| \| \| + modelWndProc | 98.63% | 0.02% |
| \| \| \| \| \| \| \| \| \| + modelLoop | 90.90% | 0.59% |
| \| \| \| \| \| \| \| \| \| \| + sub_tray | 40.67% | 23.37% |
| \| \| \| \| \| \| \| \| \| \| \| - enth | 4.48% | 4.45% |
| \| \| \| \| \| \| \| \| \| \| \| - round | 4.08% | 4.05% |
| \| \| \| \| \| \| \| \| \| \| \| - enth_to_t | 3.46% | 3.42% |
| \| \| \| \| \| \| \| \| \| \| \| - [External call] _CIexp_pentium4 | 2.26% | 2.26% |
| \| \| \| \| \| \| \| \| \| \| \| - liquid_volume | 2.16% | 2.11% |
| \| \| \| \| \| \| \| \| \| \| + subdtank | 12.60% | 5.52% |
| \| \| \| \| \| \| \| \| \| \| - flash_h | 4.11% | 0.04% |
| \| \| \| \| \| \| \| \| \| \| - round | 0.91% | 0.91% |
| \| \| \| \| \| \| \| \| \| \| - enth | 0.57% | 0.56% |
| \| \| \| \| \| \| \| \| \| \| - [External call] _CIexp_pentium4 | 0.46% | 0.46% |
| \| \| \| \| \| \| \| \| \| \| - liquid_volume | 0.41% | 0.41% |
| \| \| \| \| \| \| \| \| \| \| - enth_to_t | 0.38% | 0.38% |
| \| \| \| \| \| \| \| \| \| \| + subpv | 6.32% | 0.82% |
| \| \| \| \| \| \| \| \| \| \| - tempcalc | 5.32% | 0.36% |
| \| \| \| \| \| \| \| \| \| \| + subghexc | 6.23% | 0.20% |
| \| \| \| \| \| \| \| \| \| \| - flash_h | 5.58% | 0.05% |
| \| \| \| \| \| \| \| \| \| \| - enth_to_t | 0.23% | 0.23% |
| \| \| \| \| \| \| \| \| \| \| - [External call] _CIlog_pentium4 | 0.13% | 0.13% |
| \| \| \| \| \| \| \| \| \| \| + sub_reactor | 5.03% | 3.35% |
| \| \| \| \| \| \| \| \| \| \| - round | 0.40% | 0.39% |
| \| \| \| \| \| \| \| \| \| \| - enth | 0.34% | 0.34% |
| \| \| \| \| \| \| \| \| \| \| - [External call] _CIpow_pentium4 | 0.30% | 0.30% |
| \| \| \| \| \| \| \| \| \| \| - [External call] _CIexp_pentium4 | 0.21% | 0.21% |
| \| \| \| \| \| \| \| \| \| \| - enth_to_t | 0.21% | 0.20% |
| \| \| \| \| \| \| \| \| \| \| + sub_node | 3.80% | 1.24% |
| \| \| \| \| \| \| \| \| \| \| - flash_h | 2.07% | 0.01% |
| \| \| \| \| \| \| \| \| \| \| - enth | 0.34% | 0.34% |
| \| \| \| \| \| \| \| \| \| \| - [External call] _CIexp | 3.32% | 3.32% |
| \| \| \| \| \| \| \| \| \| \| - kmoltokg | 3.02% | 2.95% |
| \| \| \| \| \| \| \| \| \| \| - subvalvri | 2.68% | 0.94% |
| \| \| \| \| \| \| \| \| \| \| - [External call] memcpy | 2.25% | 2.25% |
| \| \| \| \| \| \| \| \| \| \| + subttank | 1.65% | 1.17% |
| \| \| \| \| \| \| \| \| \| \| - round | 0.19% | 0.19% |
| \| \| \| \| \| \| \| \| \| \| - subvalvpq | 0.87% | 0.41% |
| \| \| \| \| \| \| \| \| \| \| - [External call] _math_exit | 0.61% | 0.61% |
| \| \| \| \| \| \| \| \| \| \| - advanced_control | 0.30% | 0.01% |
| \| \| \| \| \| \| \| \| \| \| - adjust | 0.25% | 0.24% |
| \| \| \| \| \| \| \| \| \| \| - [External call] _CIsqrt | 0.19% | 0.19% |
| \| \| \| \| \| \| \| \| \| - showModelMessage | 2.68% | 0.00% |
| \| \| - modelReceiving | 0.01% | 0.00% |
| \| \| - modelSending | 0.01% | 0.00% |
| \| - [unwalkable] | 0.96% | 0.00% |

**Figure 3.** The debug call tree of original program. Some functions with low time cost have been omitted.

### 2.3. The Limit of Parallel Speedup

The speed-up ($S$) shown in Equation (4) is usually used to measure the effect of parallel acceleration, where $T(1)$ is the time used in serial and $T(p)$ is time used in parallel. According to Amdahl's law [20] as Equation (5), the upper limit of speed-up with the fixed workload is determined depending on the proportion $p$ of parallelizable parts and the number $n$ of parallelizable parts. The proportion $p$ of parallelizable parts is about 75% according to the previous analysis so the upper limit of speed-up is 4 as shown in Equation (6).

$$S = \frac{T(1)}{T(p)} \tag{4}$$

$$S = \frac{1}{1 - p + \frac{p}{n}} \tag{5}$$

$$\lim_{n \to +\infty} S = \frac{1}{1 - p} = \frac{1}{1 - 0.75} = 4 \tag{6}$$

### 2.4. Test Case

The program performance test in the following sections is carried out on a computer with AMD Ryzen™ 9 3900X with 12 cores and 64 GB RAM. As the CPU has precision boost technology, which can raise clock speed automatically, the clock speed is manually set to 2.16 GHz to prevent the clock speed change during the performance test. The compiler toolchain used is MSVC 14.28, and compilation options are the same without specified in each test. The iteration times and inter-process communication times of single system timer calls are modified to reduce the influence of system call time fluctuation. The simulation project used in the test is a nearly stable state of the ethylbenzene process and the integration time step is set to 0.125 s. The simulation time in the test is five hours, and the wall clock running time is recorded. The effect of parallel computing modification is tested on a $500 \, \text{kt} \cdot \text{a}^{-1}$ ethylbenzene process simulator.

The simulated process is based on Sinopec Research Institute of Petroleum Processing (RIPP) liquid-phase benzene alkylation, and the process flowsheet diagram is shown in Figure 4 [21]. There are two fresh feeds (benzene and ethylene) and the main reaction is benzene reacts with ethylene to produce ethylbenzene shown in Equation (7).

$$+ \quad CH_2 = CH_2 \longrightarrow \tag{7}$$

Benzene          Ethylene                    Ethylbenzene

**Figure 4.** The 500 kt · a$^{-1}$ ethylbenzene process flowsheet diagram.

## 3. Multithreading Parallelism

### 3.1. Overview of Multithreading Parallelism

Multithreading technology can execute more than one thread at the same time to improve the throughput of the program. In this study, OpenMP is used to realize parallelize multithreading.

OpenMP is a popular application programming interface providing loop-level parallelism used to code parallel threads in a shared memory system using C, C++, and FORTRAN programming languages. It uses the fork-join model of parallel execution shown in Figure 5, which provides the programmer full control to multithreading parallelism without much code changes. We achieve multithreading acceleration by producing task queues and then consume the queues in multiple threads created by OpenMP. As all threads share a common memory and the independent part is stored in private variables, the parallel algorithm can be executed on each thread and the result is stored on the shared common memory.

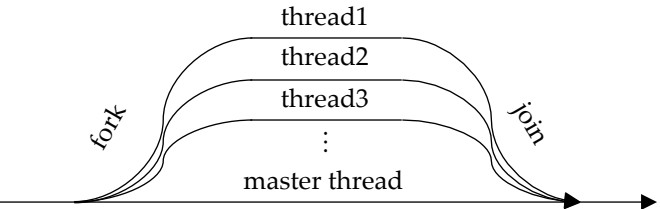

**Figure 5.** A sketch of the OpenMP fork-join model.

### 3.2. Task Allocation

All tasks run serially in the original program as shown in Figure 6. The pressure model of the column tray is simplified into algebraic equations, therefore trays of the distillation column have data dependence between each other in the same distillation column, which needs to be a serial part. The fixed bed reactor is divided into several sub-unit operations. The pressure relationship and reaction rate are also described by algebraic equations, so there is data dependence, which also needs to be a serial part.

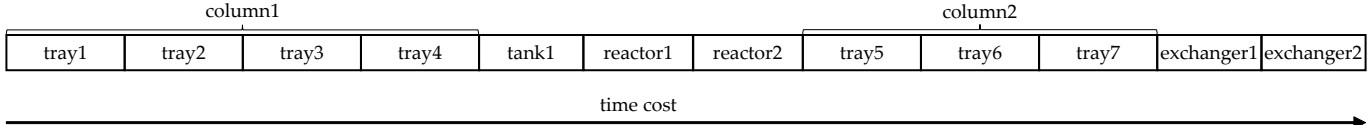

**Figure 6.** Calculation time with a single core.

Generally, the task is constructed as a directed dependency graph, then the graph is converted into a directed acyclic graph (DAG) by treating the strong components as nodes. As shown in Figure 7, the task is assigned dynamically into threads when the upper depend node finished running. However, the tasks in process dynamic simulation have a small workload so the DAG allocation method is too heavy for it. Due to the characteristics of the chemical process, the tasks have no complex dependencies, so it is possible to simplify the allocation. The allocation can be simplified by giving each task a fixed cost. The cost can be specified because dynamic simulation has a strong instantaneity, the iteration steps in each task are generally limited, and the time cost will not change greatly with working conditions. After given cost, the task allocation problem converts into balance the task queue cost with given the number of queues, which is known as parallel machines scheduling problem [22]. By balancing the task queues, it can take full use of the CPU cores without waiting shown in Figure 8.

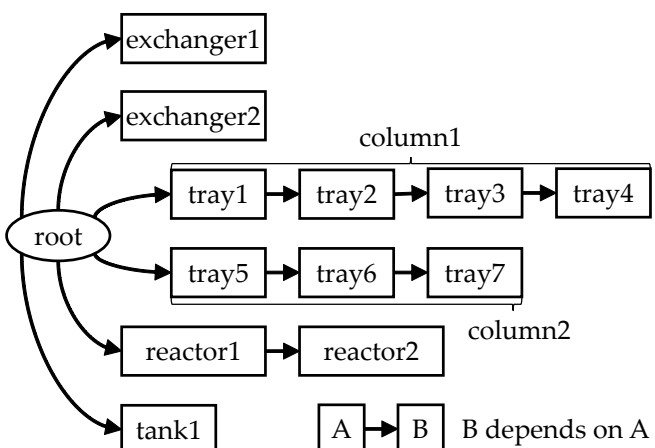

**Figure 7.** Sketch of DAG allocation.

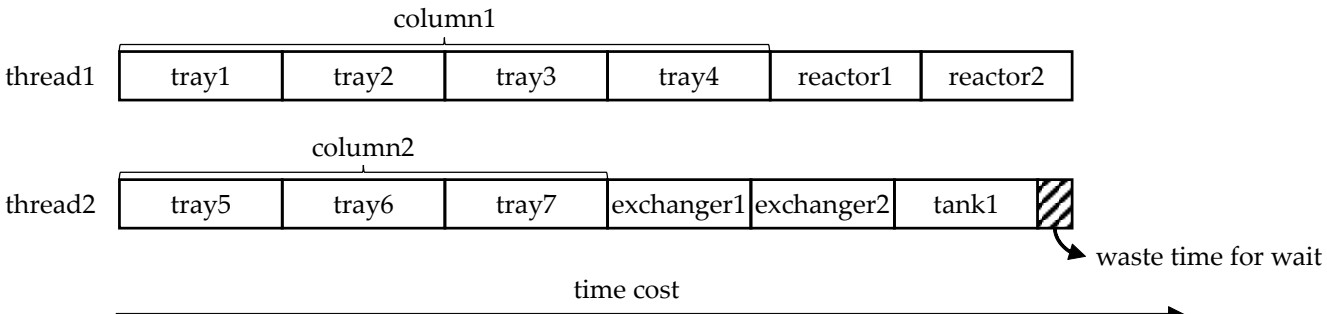

**Figure 8.** Waiting caused by imbalance task allocation.

As parallel machines scheduling problem is a non-deterministic polynomial-time hardness (NP-hard) problem, it is difficult to get the optimal solution. In order to reduce the initialization and start-up time of the whole program, we use a greedy algorithm to get a high-quality approach optimal solution to balance the task queues. The suboptimal solution is acceptable because there are often a small number of heavy tasks (like columns) and a large number of light tasks (like tanks and heat exchangers) in the chemical process, so the task allocation is generally balanced enough even if the solution is not optimal. The steps of task allocation are listed as follows:

1. Treat the tasks with dependence as a single task;
2. Sort all tasks into a list according to the time cost;
3. Pop the first task from the list and put it on the shortest task queue;
4. Sort all tasks queues according to the time cost;
5. If there is any remaining task in the list, return to Step 3. Otherwise, end of the task allocation.

Because the compiler only supports OpenMP 2.0 [23], the task queue is consumed in the loop. The principle of multithreading rewriting is shown in Algorithm 1.

---

**Algorithm 1** Pseudocode for task queue consuming

---

#pragma omp parallel for num_threads($n$) schedule(static)
**for** $i = 0$ to $n$ **do**
    consume(task queue[$i$])
**end for**

---

*3.3. Simulation Performance Results*

The running speed comparison between the original program and the multithreaded program is shown in Table 2. The enhanced instruction is turned off in this test.

**Table 2.** The performance results of multithreading parallelism.

| Thread Number | Wall Clock Time/ms | Speed Up | Simulation Time / Wall Clock Time |
|:---:|:---:|:---:|:---:|
| 1 | 185,378 | - | 97.10 |
| 2 | 117,268 | 1.58 | 153.49 |
| 3 | 100,378 | 1.85 | 179.32 |
| 4 | 89,572 | 2.07 | 200.96 |
| 5 | 90,293 | 2.05 | 199.35 |
| 6 | 90,590 | 2.05 | 198.70 |

It can be observed that when the number of parallels is $n = 2$, the speed-up is closest to the theoretical speed-up limit. When the number of parallels is $n > 4$, the effect of additionally increasing the number of parallels is not significant. This may be because there are two distillation columns that have more trays than other columns in the ethylbenzene synthesis process unit, and the calculation time of these two column modules is significantly longer than that of other units. Therefore, when $n = 2$, the task allocation is more balanced, and the overhead of multithreading parallel waiting is less. After $n = 4$, as shown in Figures 8 and 9, due to the imbalance of task allocation, some threads are waiting. Therefore, increasing the number of parallels $n$ will make more deviation from the theoretical speed-up limit.

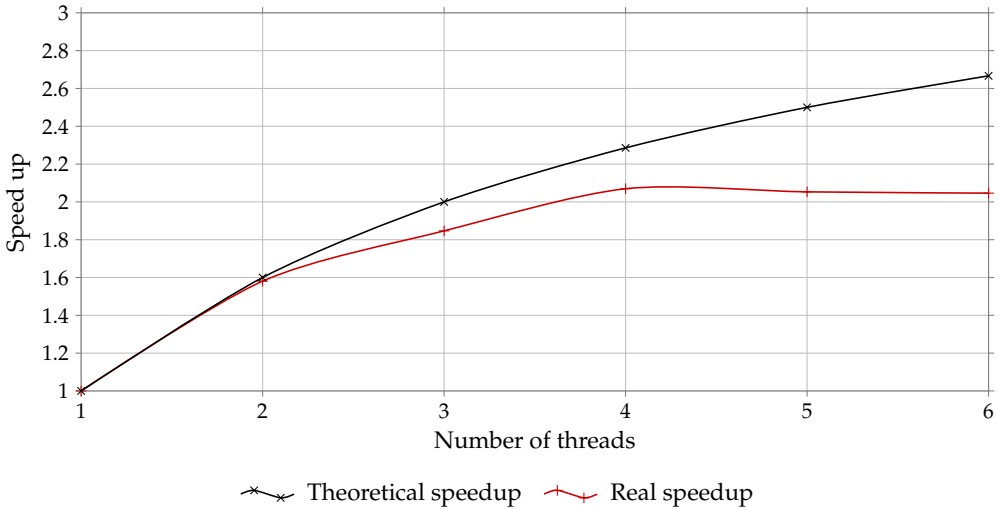

**Figure 9.** Time cost with different number of threads.

## 4. Vectorization Parallelism

### 4.1. Overview of Vectorization Parallelism

The single instruction multiple data (SIMD) technology of the processor is used in the parallelism at data-level. The same operation can be performed on a set of data in one instruction at one moment [24]. Modern compilers can provide a compiler automatic vectorization (CAV) capability to automatically expand the loop to data-level parallelism in the process of compilation, but it cannot perform vector parallelism automatically for more complex loops or modify the data structure to fit vectorized parallelism. Intel intrinsics inline functions are used to rewrite the loop that cannot perform vector parallelism automatically that reported by the compiler [25].

### 4.2. Vectorization Parallelism Example

Because there are the same types of calculation for each component in the thermodynamic calculation, but there are generally complex logic branches in the loop, it can be vectorized manually. For example, the original pseudocode of flash part calculation is shown in Algorithm 2. It performs the same operation on each component which is suitable for vectorization. However, it cannot be vectorized automatically because of the conditional branches in the loop. Therefore, it needs to be vectorized manually. To fit the vectorized parallelism, the data structure of thermodynamic parameters has changed from array of structures (AoS) shown in Figure 10a to structure of arrays (SoA) shown in Figure 10b [26]. First, remove the conditional branch from the loop, and then use the inline function to manually vectorize the actual calculation line, the manually vectorized pseudocode is shown in Algorithm 3.

With the support of the AVX2 instruction set, the processor packages multiple data into a vectorized element. During calculation, vectorized elements are directly processed, and four double-precision floating-point numbers can be calculated together at one moment. The effect is shown in Figure 11. Theoretically, this vectorization method can be up to four times faster than the original calculation.

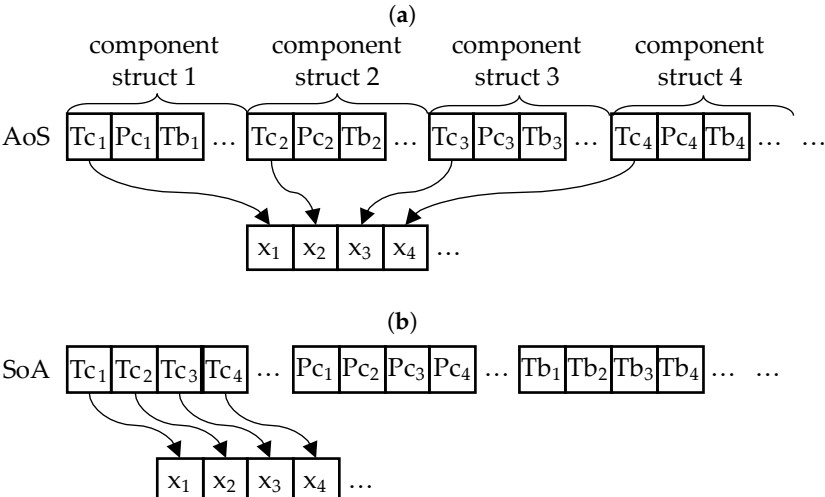

**Figure 10.** The data structure of the thermodynamic property. (**a**) Array of structures. (**b**) Structure of arrays.

---

**Algorithm 2** Original pseudocode for solving flashing problems

---

**for all** components **do**
  **if** $z_i > 0$ **then**
    $k_i = Pc_i/P * \exp(hc_i * (1 - Tc_i/T_0))$
  **else**
    $k_i = 0$
  **end if**
**end for**

---

**Algorithm 3** SIMD vectorize paralleled pseudocode for solving flashing problems

---

**for all** groups of components **do**
  $k_{i/4} = \mathrm{mul\_pd}(\mathrm{div\_pd}(Pc_{i/4}, P), \exp\_pd(\mathrm{mul\_pd}(hc_{i/4}, \mathrm{sub\_pd}(1, \mathrm{div\_pd}(Tc_{i/4}, T_0)))))$
**end for**
**for all** components **do**
  **if** $z_i > 0$ **then**
    $k_i = 0$
  **end if**
**end for**

---

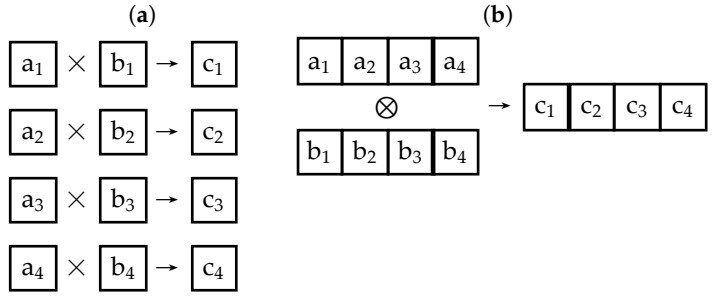

**Figure 11.** Difference between operation with normal instruction and SIMD instruction. (**a**) Normal instruction. (**b**) SIMD instruction.

*4.3. Simulation Performance Results*

The running speed comparison between the original program and the vectorized program is shown in Table 3. In this test, the multi-threading parallelism is not turned on.

**Table 3.** The performance results of vectorization parallelism.

| Vectorization Way | Wall Clock Time/ms | Speed Up | Simulation Time / Wall Clock Time |
|---|---|---|---|
| none | 185,378 | - | 97.10 |
| CAV | 145,444 | 1.27 | 123.76 |
| manual | 138,311 | 1.34 | 130.14 |

It can be observed that there is an acceleration effect after automatic vectorization. According to the observation in the debugging mode, the main acceleration effect happens in calculating the natural logarithm.

Based on automatic vectorization, several thermodynamic calculation functions with the most calls are found. Manual vectorization rewriting is carried out for them. The speed-up effect of manual vectorized parallelism is shown in Table 3. Manual vectorization can further improve the solving speed based on compiler automatic vectorization, but the speed increase is not significant because the total time cost of the manually modified function is not long enough.

## 5. Combining Multithreading and Vectorization Parallelism

*5.1. Acceleration Effect*

The vectorization parallelism can provide a higher speed-up in a different way from multithreading parallelism. Thus, these two methods can be used at the same time. The acceleration effect is shown in Table 4. The acceleration effect of vectorization parallelism is not as significant as multithreading parallelism because it is small granularity parallelism.

**Table 4.** The performance results of combining parallelism.

| Parallelism Method | Wall Clock Time/ms | Speed Up | Simulation Time / Wall Clock Time |
|---|---|---|---|
| none | 185,378 | - | 97.10 |
| multithreading | 89,572 | 2.07 | 200.96 |
| vectorization | 138,311 | 1.34 | 130.14 |
| combining | 70,980 | 2.61 | 253.59 |

*5.2. Impact on Simulation Results*

Luyben [27] reports the snowball effect in ethylbenzene process, which causes the recycle flow rate to be sensitive to small changes in the process. This means recycle flow rate is a good indicator of solution stability of the simulator. We apply a step signal to the set point of the level controller of the last reflux drum (D-4) to observe the variation of recycle flowrate with different acceleration methods. The simulator program output shown in Figure 12 are almost identical, which proves that the solution is not impacted by each acceleration methods.

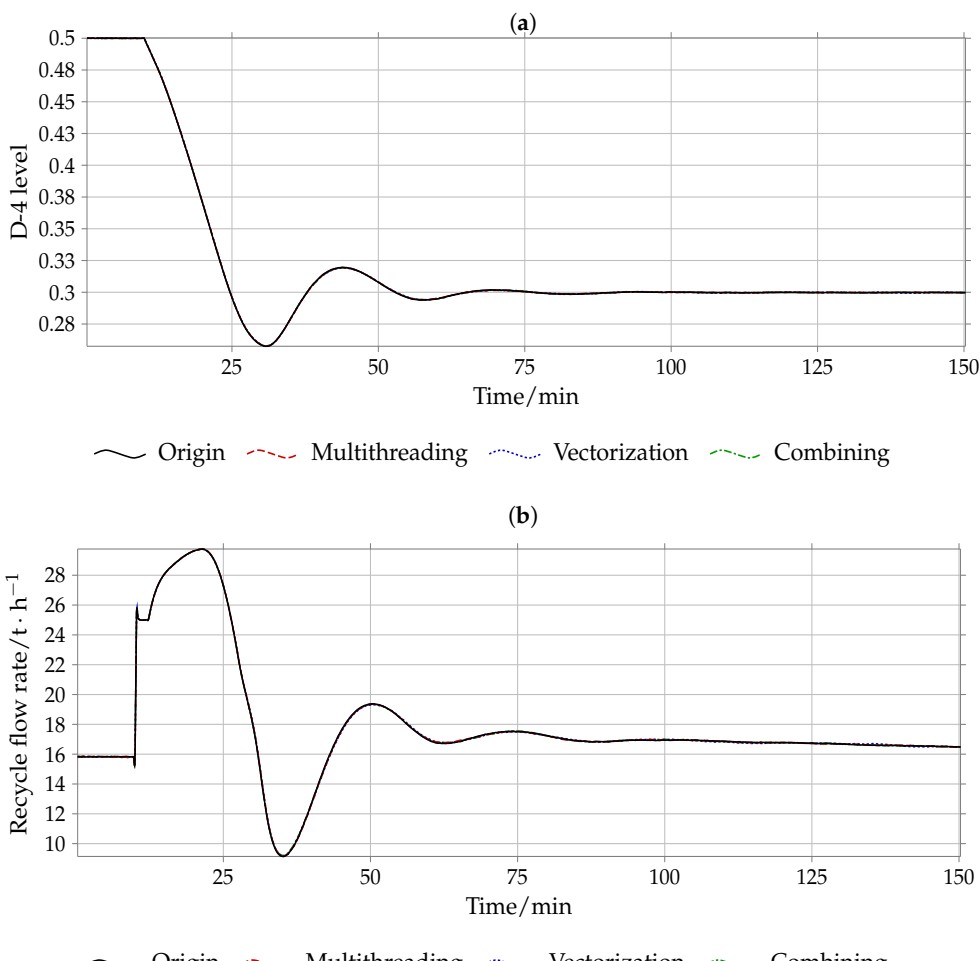

**Figure 12.** The simulation result with different acceleration methods. (**a**) D-4 level. (**b**) Recycle flow rate.

## 6. Conclusions

In order to improve the speed of dynamic simulation to fit the current needs of increasing complexity in chemical processes and data-driven methods, a parallel scheme for dynamic simulation is designed based on the calculation and solving characteristics of dynamic simulation. The parallel part separated by unit operation is more in accord with the laws of process. With a $500 \text{ kt} \cdot \text{a}^{-1}$ ethylbenzene process test case, the theoretical analysis shows that the speed-up limit is 4. The multithreading parallel is used first, parallel machines scheduling problem solved by greedy algorithm is used to replace the dependence analysis by DAG. The results of the multithreading parallel show that the number of parallel cores is not the more the better because the distillation column will be a serial control step. The highest efficiency appears in $n = 4$ in our test case. In addition, we also used vectorized parallelism. The CAV is not enough to make full use of the CPU performance. By manually rewriting part of the hot functions and changing the data structure to the SoA, the parallel speed-up effect can be further improved.

Through these two kinds of parallel methods, the efficiency of the simulator can be effectively improved without affecting the results, which can be increased to 261% of the original program, and the simulation speed can reach 253.59 times of the real-time speed, which can better meet the various needs of the simulator.

**Author Contributions:** Conceptualization, J.Z.; Investigation, Z.L.; Methodology, J.Z.; Project administration, Z.D.; Resources, Z.D. and J.W.; Software, J.Z.; Supervision, Z.D.; Validation, Z.L.; Writing—original draft, J.Z. and Z.D. All authors have read and agreed to the published version of the manuscript.

**Funding:** This research received no external funding.

**Institutional Review Board Statement:** Not applicable.

**Informed Consent Statement:** Not applicable.

**Data Availability Statement:** Data sharing not applicable.

**Acknowledgments:** Thanks Chao Song for giving us help in writing and Peiran Yao from University of Alberta giving us a lot of help in computer programming.

**Conflicts of Interest:** The authors declare no conflicts of interest.

## Abbreviations

The following abbreviations are used in this manuscript:

| | |
|---|---|
| AVX2 | Advanced vector extensions 2 |
| CAV | Compiler automatic vectorization |
| DAE | Differential-algebraic equations |
| DAG | Directed acyclic graph |
| DSO | General dynamic simulation & optimization system |
| KKT | Karush–Kuhn–Tucker |
| NP-hard | Non-deterministic polynomial-time hardness |
| SIMD | Single instruction multiple data |
| TEP | Tennessee Eastman process |

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
