# Peer review of "A Parallel Processing Approach to Dynamic Simulation of Ethylbenzene Process"

_processes, doi:10.3390/pr9081386_

Round 1
Reviewer 1 Report
This manuscript deals with a parallel dynamic simulation for a 500 kt∙a−1 ethylbenzene process taking into account the character of process and the development of computer, and multithreading and vectorization parallel computing modifications were carried out based on General Dynamic Simulation and Optimization System (DSO) using the all features of modern CPUs. In addition, the high-level multithreading parallelism and assign tasks were applied according to unit operations which provided clearer task assign and lower communication costs. Since the computer aided simulations have high impact on the chemical industrial processes, this investigation deserves attention.
The manuscript is fairly well written and the results are clearly presented. However, some minor revisions should be made before publication:
1) p. 3 line 101 Figure 2b is no mentioned in the text of manuscript. Please, check its role in your explanations.
2) p. 6 line 165 The authors write that “The effect of parallel computing modification is tested on a 500 kt・a−1 ethylbenzene process simulator, the process is shown in Figure 4 [21].”, but it is not clear what the technology is exactly (e.g. alkylation of benzene with ethylene or other one). It would be more advantageous if you inserted a reaction equation about this chemical process.
3) ps. 13–14 Style of the references does not meet the requirements of journal Processes. For example, in case of “Computers & Chemical Engineering, Chemie Ingenieur Technik, AIChE Journal, Industrial & Engineering Chemistry Research” you should use them in these forms: Comput. Chem. Eng., Chem. Ing. Tech., AIChE J., Ind. Eng. Chem. Res. Please, check all of them and modify them accordingly.
4) There are some typing or grammatical errors in the text:
p. 2 line 73 “… Moore’s Law …” instead of Moore’s law
p. 3 line 82 “… a parallel dynamic simulation which take the character of process and the development of computer into account.” instead of a parallel dynamic simulation taking into account the character of process and the development of computer.
p. 6 line 150 “… Amdahl’s Law …” instead of Amdahl’s law
line 157 “… AMD ryzen™9 3900X ...” instead of AMD Ryzen™ 9 3900X with 12 cores
Author Response
1) p. 3 line 101 Figure 2b is no mentioned in the text of manuscript. Please, check its role in your explanations.
Reply:
Thank you for your pointing out. We have revised the manuscript, in which we mentioned Figure 2b and gave more explanation in Line 141-143.
2) p. 6 line 165 The authors write that “The effect of parallel computing modification is tested on a 500 kt・a−1 ethylbenzene process simulator, the process is shown in Figure 4 [21].”, but it is not clear what the technology is exactly (e.g. alkylation of benzene with ethylene or other one). It would be more advantageous if you inserted a reaction equation about this chemical process.
Reply:
We have added the chemical reaction in ethylbenzene process, as shown in Equation 7.
3) ps. 13–14 Style of the references does not meet the requirements of journal Processes. For example, in case of “Computers & Chemical Engineering, Chemie Ingenieur Technik, AIChE Journal, Industrial & Engineering Chemistry Research” you should use them in these forms: Comput. Chem. Eng., Chem. Ing. Tech., AIChE J., Ind. Eng. Chem. Res. Please, check all of them and modify them accordingly.
Reply:
Thank you for your reminder. We have checked the reference style and corrected them.
4) There are some typing or grammatical errors in the text:
- 2 line 73 “… Moore’s Law …” instead of Moore’s law
- 3 line 82 “… a parallel dynamic simulation which take the character of process and the development of computer into account.” instead of a parallel dynamic simulation taking into account the character of process and the development of computer.
- 6 line 150 “… Amdahl’s Law …” instead of Amdahl’s law
line 157 “… AMD ryzen™9 3900X ...” instead of AMD Ryzen™ 9 3900X with 12 cores
Reply:
Thank you for your suggestion. We have corrected the typing and grammatical errors in this revised manuscript. All the modifications have been highlighted in the attachment diff.pdf .
Thank you again for your valuable comments on our manuscript.

Reviewer 2 Report
The English needs a lot of minor changes. I will leave a copy with the editors with the main issues highlighted. In nearly all the places where you use the word "solving" - this is a verb and you are using it as a noun. You need to change 95% of these to "solution" and rephrase. A good example of this is on Line 98 "the actual solving form is as Eqn 3" - this would be rephrased as "the actual solution is given by Eqn 3".
Figure 2 - the difference between a and b is not very clear. Is it simply the outline around "Unit Operation"?
Figure 4 - Drawing quality could be improved - recommend using different equipment weight for equipment and pipes. Significant control valves should be shown in this figure.
You mention in both the abstract and conclusions that the solution can be accelerated so that it can solve the model 250 times faster than the actual process time. However, you need to make a clearer explanation about why the actual process time is important - would it matter if the model ran slower than the process. How much faster does the DSO need to be to be useful?
Please provide some evidence that the solution is not impacted by each of the algorithms that you employ.

Author Response
1) The English needs a lot of minor changes. I will leave a copy with the editors with the main issues highlighted. In nearly all the places where you use the word "solving" - this is a verb and you are using it as a noun. You need to change 95% of these to "solution" and rephrase. A good example of this is on Line 98 "the actual solving form is as Eqn 3" - this would be rephrased as "the actual solution is given by Eqn 3".
Reply:
Thank you for your comments. When we used the word "solving", we actually wanted to emphasize the solving process rather than the way or the answer as emphasized in "solution", so thank you for your comment that made us realize the error in our wording. We have corrected the language mistakes and polished the language carefully. I hope the current quality could reach your standard.
2) Figure 2 - the difference between a and b is not very clear. Is it simply the outline around "Unit Operation"?
Reply:
Thank you for your question. We want to describe the change of the whole solution process in multithreaded modification, in which only unit operations are paralleled. So the unit operations are illustrated as the overlapping form in Figure 2b. We have adjusted the picture to make it easier to understand.
3) Figure 4 - Drawing quality could be improved - recommend using different equipment weight for equipment and pipes. Significant control valves should be shown in this figure.
Reply:
We appreciate your valuable suggestion. We have reorganized this process flowsheet diagram.
4) You mention in both the abstract and conclusions that the solution can be accelerated so that it can solve the model 250 times faster than the actual process time. However, you need to make a clearer explanation about why the actual process time is important - would it matter if the model ran slower than the process. How much faster does the DSO need to be to be useful?
Reply:
Thank you for your question about the actual process time. The simulator can only be applied combining with real processes if it is faster than the real process. We introduced the application of the solver in the introduction in the previous manuscript, and added some description of the significance of improving the speed of the solver faster than the real process in the new manuscript. The many demands of the simulator require faster speeds such as MPC, RTO, RL, OTS, etc. But the existing plantwide level simulators are not fast enough to meet the application requirements. The application of the simulator requires it to be as fast as possible to leave sufficient running time for optimization, control, interaction, etc. We believe that making a simulator faster is a justified need without more explanation than our existing manuscripts.
Thank you very much for your suggestions!
5) Please provide some evidence that the solution is not impacted by each of the algorithms that you employ.
Reply:
We tested the results with different acceleration methods and added a new section 5.2 to clarify the relevant facts. It could be clear that the solution is not impacted by each acceleration methods.
Thank you again for your valuable comments on our manuscript.
